# Rainwater Harvesting and Treatment: State of the Art and Perspectives

**Anita Raimondi** [1], **Ruth Quinn** [2,*], **Gopinathan R. Abhijith** [3], **Gianfranco Becciu** [1] **and Avi Ostfeld** [4]

1. Department of Civil and Environmental Engineering, Politecnico di Milano, 20133 Milano, Italy
2. Department of Civil Engineering and Construction Studies, Atlantic Technological University Sligo, F91 YW50 Sligo, Ireland
3. Department of Civil Engineering, BITS Pilani Hyderabad Campus, Hyderabad 500078, India; abhijith.gr@hyderabad.bits-pilani.ac.in
4. Faculty of Civil and Environmental Engineering, Technion—Israel Institute of Technology, Haifa 32000, Israel
* Correspondence: ruth.quinn@atu.ie

**Abstract:** Rainwater harvesting is an ancient practice currently used for flood and drought risk mitigation. It is a well-known solution with different levels of advanced technology associated with it. This study is aimed at reviewing the state of the art with regards to rainwater harvesting, treatment, and management. It focuses on the environmental and social benefits of rainwater harvesting and links them to the Sustainable Development Goals. The review identifies characteristics of laws and regulations that encourage this practice and their current limitations. It presents methodologies to design a rainwater harvesting system, describes the influence of design variables, and the impact of temporal and spatial scales on the system's performance. The manuscript also analyzes the most advanced technologies for rainwater treatment, providing insights into various processes by discussing diverse physiochemical and biological technology options that are in the early stages of development. Finally, it introduces trends and perspectives which serve to increase rainwater harvesting, water reuse, and effective management.

**Keywords:** rainwater harvesting; climate change; rainwater harvesting systems; rainwater treatment; low impact development; sustainable urban drainage systems; water supply; rainwater reuse; mitigation; management

## 1. Introduction

Water is strongly related to human health, socio-economic prosperity, food production, and the environment. The water–food–energy nexus identifies this natural resource as fundamental for life on Earth. Despite this, millions of people in developing countries still do not have access to enough clean water to satisfy basic needs.

The sixth Sustainable Development Goal (SDG) of the United Nations Agenda 2030, Clean Water and Sanitation, states that more than 733 million people still live in countries with high and critical levels of water stress [1].

The world population is growing (especially in developing countries), and by 2050 about 64% of people are expected to live in cities. This will cause an increase in water demand, which has already quadrupled in the 20th century [2].

Moreover, climate change is intensifying extreme events all around the world. This occurs not only in countries traditionally affected by water scarcity but also in regions usually characterized by high availability of water resources, which are often misused or wasted. The World Meteorological Organization (WMO) states that droughts have risen by 29% since 2000, and 2.3 billion people suffered from water stress in 2022, forecasting that droughts may affect over three-quarters of the world population by 2050 [3]. At the same

time, in the past two decades, 163 annual floods were recorded, and 223 large-scale floods occurred in 2021 alone [4].

Rainwater harvesting (RWH) and reuse is an ancient water supply practice; examples of systems date from the Neolithic period [5]. The development of civilizations often benefited from the storage of rainwater and its planned use over time. RWH is still utilized as the primary source of water supply for millions in developing countries [6]. However, even in developed countries, rainwater harvesting and reuse are increasingly encouraged by regulations and laws, representing a sustainable solution for improving water supply resilience.

RWH and reuse belong to a set of water management techniques known as best management practice (BMP), which is also named low-impact development (LID) solutions or sustainable drainage systems (SuDS) depending on the country.

The interest in this practice is evident by the sharp increase in the number of documents obtained through a Scopus search with the keyword "rainwater harvesting", which shows no signs of diminishing (Figure 1).

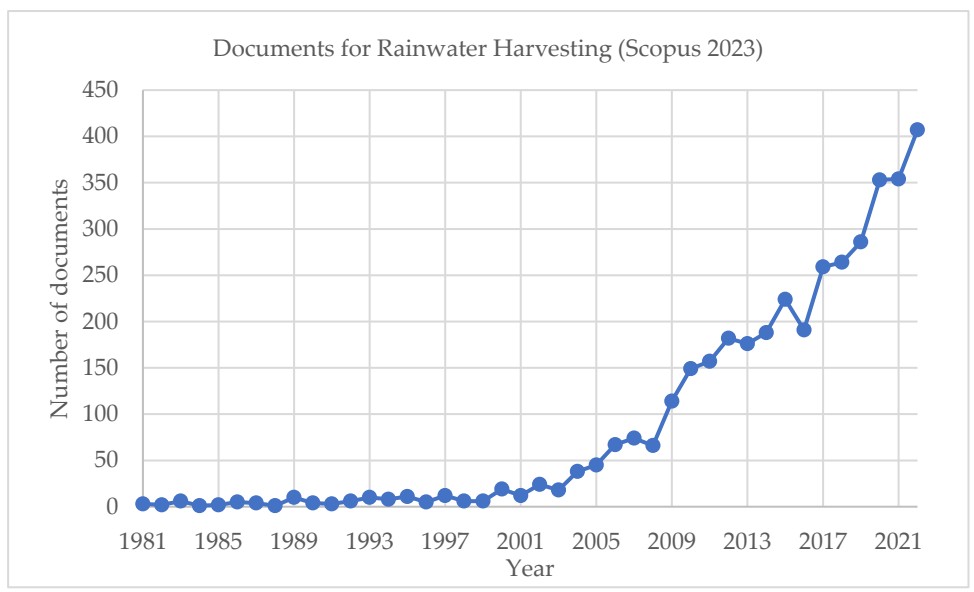

**Figure 1.** Number of documents in Scopus (6 February 2023) corresponding to the keyword "rainwater harvesting".

The interest in this practice is widespread in all continents (Figure 2), with a high number of contributions from the United States, China, and India. These data confirm global interest in this topic and in efforts towards making cities and communities resilient to challenges posed by climate change.

The top journals publishing papers on RWH are *Water* (MDPI), *Agricultural Water Management* (Science Direct), *Journal of Cleaner Production* (Science Direct), *Water Resources Management* (Springer), *Resources Conservation and Recycling* (Science Direct), *Water Science and Technology Water Supply* (IWA Publishing), *Science of the Total Environment* (Elsevier), *Sustainability* (MDPI), and *Physics and Chemistry of the Earth* (Science Direct).

This manuscript aims to review papers on RWH and treatment and identify the most important findings and progress in this field of research.

Section 2, "Rainwater Harvesting", introduces the multiple advantages of this practice and its technical, social, and financial limitations (Section 2.1) and links them to the SDGs, providing new insights into the utility of these systems. Section 2.2 presents key characteristics of regulations, laws, and design manuals which can help or hinder widescale RWH implementation.

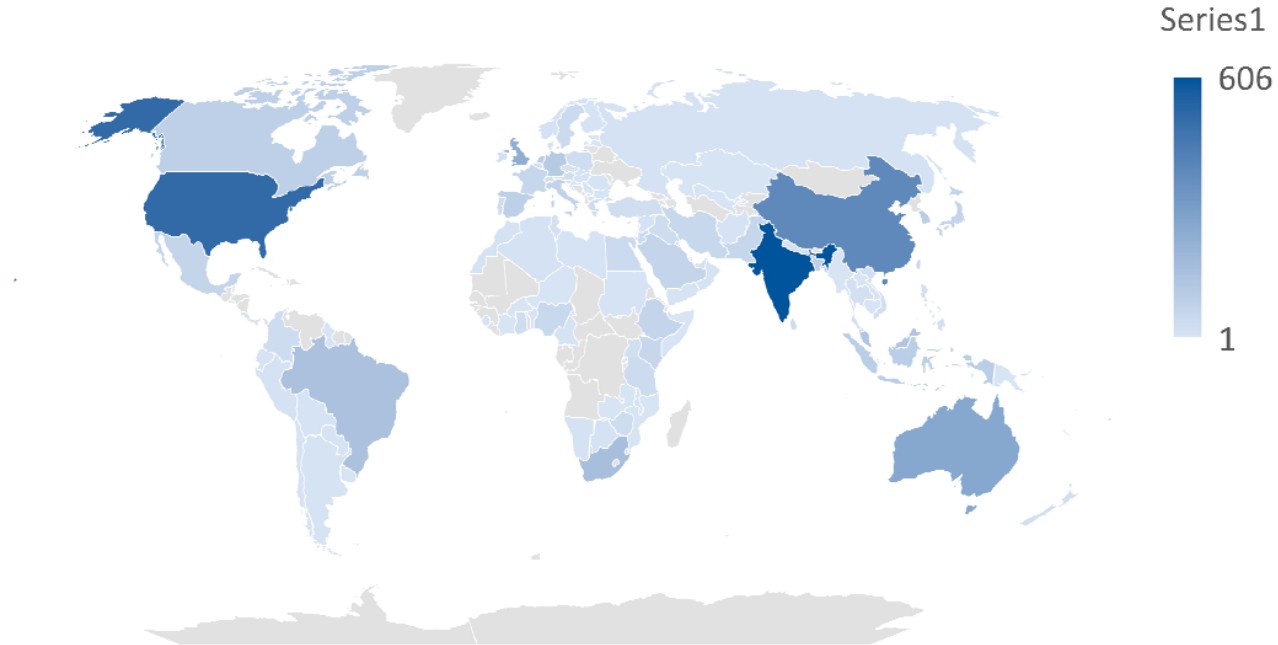

© Australian Bureau of Statistics, GeoNames, Geospatial Data Edit, Microsoft, Navinfo, OpenStreetMap, TomTom, Wikipedia

**Figure 2.** World distribution of manuscripts corresponding to the keyword "rainwater harvesting" (Scopus search, 6 February 2023).

Section 3, "Rainwater Harvesting Systems", describes modern RWH systems focusing on domestic usage. It reviews state-of-the-art methods for their design and modeling. It also examines the performance of these systems depending on different variables as well as their spatial and temporal scales, and the aim of the RWH system.

Section 4, "Rainwater Treatment", presents current and emerging technologies to achieve set water quality standards for the different reuses of rainwater.

## 2. Rainwater Harvesting

### 2.1. Advantages and Limitations

The environmental benefits of RWH in urban settings are well understood, with numerous papers dedicated to describing their ability to mitigate pluvial flooding, provide an additional water supply and even decrease greenhouse gas emissions [7,8]. These systems are part of a family of holistic water management approaches known as SuDS, which aim to manage stormwater at or close to its source, mimic natural drainage and encourage runoff infiltration, attenuation, and passive treatment. Thus, their overall sustainability should be examined when discussing the advantages of these systems instead of focusing on individual benefits. The word sustainability is commonly used in literature and academia, but it remains an ambiguous concept with countless interpretations, many of which are based on context-specific understandings [9]. The most prevalent description incorporates three interrelated or interlocking 'pillars' containing economic, social, and environmental factors or aims [9]. Until the benefits of RWH have been fully articulated within all of these spheres, it is unlikely that these systems will reach widespread implementation. Indeed Campisano et al. [7] identified the need for extensive further research in both financial modelling (economic pillar) and community engagement (social pillar). These pillars, as with the concept of 'sustainability', do not provide specific goals to achieve these targets, and although they are helpful as a framing mechanism for discussion; their use as a framework to illustrate the benefits of RWH is limited. The United Nation's more diverse set of SDGs, which have the three pillars embedded in their formulation, provides a much more useful tool [10].

De Sá Silva et al. [8] examine RWH within the context of three sustainability pillars and the SDGs but limit their analysis of the goals of clean water (Goal 6), affordable and clean energy (Goal 7), sustainable cities and communities (Goal 11), and actions to avoid the impacts of climate change (Goal 13). This is a small subset of the goals, and the potential beneficial effects of RWH are more significant and can impact a far larger number of the SDGs, as shown using selected examples in Table 1 below.

**Table 1.** Beneficial impacts of RWH on the achievement of the Sustainable Development Goals.

| Sustainable Development Goal | Associated Sustainability Pillar [11] | Rainwater Harvesting Advantages |
| --- | --- | --- |
| 1. No Poverty | Economic | Girma et al. [12] showed that the use of integrated RWH practices has a significant negative impact on the probability that a household is multidimensionally poor in Ethiopia. |
| 2. Zero Hunger | Economic | Kelemewerk Mekuria et al. [13] demonstrated that adopting RWH technology has a positive and significant effect on the livelihood of farmers in terms of household annual farm income and food security in Ethiopia. |
| 3. Good Health and Wellbeing | Economic | RWH can provide an additional source of potable water, improving hygiene and thus decreasing disease prevalence. For example, Fry et al. [14] examined 37 West African cities and estimated that domestic RWH with 400 L storage capacity could result in a 9% reduction in disability-affected life years. |
| 4. Quality Education | Social | Graham et al. [15] summarized the burden of water collection in Sub-Saharan Africa that usually falls on women and children, negatively impacting their school attendance and performance. Mwenge Kahinda et al. [16] described one of the main advantages of RWH as alleviating the burden of having to travel great distances to fetch water. |
| 5. Gender Equality | Social | In addition to minimizing the distance travelled and time taken to fetch water, as discussed above, RWH also increases hygiene provisions in schools, decreasing the educational time girls miss due to menstruation [17]. |
| 6. Clean Water and Sanitation | Economic | Both Campisano et al. [7] and de Sá Silva et al. [8] summarized numerous studies which show that RWH is an excellent source of additional water supply. |
| 7. Affordable and Clean Energy | Economic | De Sá Silva et al. [8] summarized key benefits in this area regarding the ability of RWH to minimize the energy needed to treat drinking and wastewater. |
| 8. Good Jobs and Economic Growth | Economic | In certain circumstances, RWH can offer a cheaper water supply than alternatives such as street vendors [18]. Jobs can also be created by policies which support RWH as they create a demand for associated products [19]. |
| 9. Industry, Innovation, and Infrastructure | Economic | RWH can delay the need to upgrade existing water treatment plants [20] and reduce the burden on combined sewer systems [20]. |
| 10. Reduced Inequalities | Social | RWH can reduce inequalities by providing a clean decentralizing water supply [21]. |
| 11. Sustainable Cities and Communities | Environment | RWH can be incorporated into cities' approaches to governance and offers an opportunity to increase the sustainability of municipalities by providing a decentralized supplemental water supply and increasing resilience to flooding [22]. |

**Table 1.** *Cont.*

| Sustainable Development Goal | Associated Sustainability Pillar [11] | Rainwater Harvesting Advantages |
|---|---|---|
| 12. Responsible Consumption and Production | Environment | Modelling of RWH systems showed that RWH has the potential to reduce the amount of detergent used in clothes washing as they supply soft water, which requires fewer additives to clean garments [23]. |
| 13. Climate Action | Environment | RWH strengthens resilience and adaptive capacity to climate-related disasters such as droughts and intense rainfall [7,8]. |
| 14. Life Below Water | Environment | RWH systems, if implemented correctly, can help to prevent combined sewer overflow, which, if left unchecked, can cause a detrimental impact on aquatic environments [24]. |
| 15. Life on Land | Environment | In drylands, agricultural schemes which harvest water enhanced local arthropod abundance [25]. RWH has been shown to improve the soil nutrient profile, increasing biomass production and thus supporting higher numbers of plants and animals [26]. |
| 16. Peace, Justice, and Strong Institutions | Social | Integrated water resources management, including RWH, offers communities a chance to engage in water management planning and decision making; bringing people together to discuss water issues can also reduce violence [27]. Communities can also join in the design and operation of these systems by emptying them in advance of a large rainfall event [28]. |
| 17. Partnership for the Goals | Social | An extensive global partnership focused on the development of RWH is absent. However, numerous examples of international collaborations exist, such as between Brazil and China [29]. |

The use of rainwater harvesting for water supply is not exclusive to developing countries; it is also the primary water source in rural areas of Australia (especially in the South) and in semi-arid areas of the USA [30]. It is essential to critically analyze all benefits of RWH and understand its limitations to avoid unintended consequences. For example, RWH is often proposed as a viable alternative to water sources contaminated with arsenic or high salinity [31]. However, Naser et al. [32] examined the potential adverse health impacts of relying on rainwater as a primary water source, including higher cardiovascular mortality and morbidity associated with consuming water low in calcium and magnesium salts [32]. Therefore, they urge caution when promoting RWH systems and suggest further research into mixing groundwater and rainwater, re-mineralizing rainwater and food fortification with calcium and magnesium [32]. In addition, RWH does not provide a complete solution to water management problems. For example, Ghodsi et al. [24] demonstrated that cisterns could only decrease combined sewer overflow by 8–18% in Buffalo, New York. In terms of water supply, numerous studies across different climates demonstrated a non-potable water supply efficiency (proportion of water demand supplied by RWH) below one [33–35]. So, although RWH is a powerful tool for reducing the need for municipal water, they cannot reasonably supply enough water on their own. In addition, performance is highly dependent on rainfall, making RWH highly vulnerable to climate change. Results from a study by Musayev et al. [36] indicated that climate change will have little impact on rainwater harvesting. Still, care needs to be taken to ensure that RWH is optimally designed to consider changes in rainfall patterns. Finally, to design RWH systems to achieve the larger-scale environmental benefits mentioned above, specialist modelling software is needed; often, SWMM is used, which is open-source and thus available to everyone [24].

Again, however, specialist skills are required to design the scenarios and models, and the calculations are often computationally extensive, requiring a moderate to high-specification computer, which may be beyond the reach of some.

### 2.2. Regulations and Laws

Regulations can both help and hinder the widespread adoption of RWH systems. Campisano et al. [7] reviewed the implementation of RWH and regulations globally and identified Germany as a leader in promoting the widespread use of RWH for domestic use. Germany's success is a direct result of the financial promotion (by grants and subsidies) of household RWH at the local government level. Schuetze [37] explained that their approach focuses on saving water, flood control, and protecting existing infrastructure and ecosystems. Other countries, such as the UK, adopt standards on a national scale [38]. In contrast, in larger countries such as Brazil and the USA, regulations are implemented at lower governmental levels [39,40]. In addition, some guidance and regulation focus on a particular environmental issue. For example, the British Standard [38] on RWH provides guidance on the design, installation, and maintenance of systems purely for non-potable water supply. Furthermore, some countries allow rainwater harvesting to be used as a source of drinking water, but some prohibit it requiring separate plumbing systems for potable and harvested water to prevent contamination [7]. Although the installation of RWH is largely optional, some areas, such as regions of New Zealand, have made it compulsory, which impacts how people interact with these systems [41]. For example, Gabe et al. [41] found that participants' satisfaction with a mandatory RWH scheme in Auckland increased over time, indicating that long-term usage increases the recognition of RWH benefits. In summary, the most successful regulations and support, in terms of ensuring the widespread implementation of RWH, deal with not only the environmental benefits of these systems but also provide financial incentives [7]. Cost is often identified as a barrier to RWH installation, so addressing this is a natural step towards increasing their popularity [42]. In future, regulations which recognize the costs of social learning, as identified by Gabe et al. [41], are likely to be more beneficial in terms of the sustainability of these systems in the long term. Finally, when measuring the long-term success of these regulations, it is crucial to examine not only the number, as regulations regarding regular maintenance may necessitate the removal or decommissioning of systems earlier than initially planned.

## 3. Rainwater Harvesting Systems

RWH represents, together with wastewater reuse and desalination, one of the more popular technological solutions to address water scarcity, which is increasing in many areas due to climate change. The construction of large-scale projects such as dams, pumping stations, and long-distance pipelines often involves socio-economic and environmental impacts and demands high investments. Alternatively, domestic RWH is locally feasible; obviously, storage volumes are limited to catchment surfaces. However, widespread implementation of this practice in urban areas can contribute to meeting multiple water management objectives [43].

RWH provides several benefits to the local environment. It can provide a non-potable water source, reducing demand from distribution systems. Moreover, it mitigates urban floods, reduces water pollution (reducing the activation of combined sewer overflows) and preserves high-quality water sources.

The different targets can sometimes be opposed; for example, for flood mitigation targets, some regulations suggest emptying the tank within 48 h to have the storage capacity fully available when rainfall occurs [44]. Conversely, for the water supply target, during dry periods, the water in the tank is needed for use and emptying it prematurely can result in water scarcity. To overcome this limit, a portioning of the storage capacity can be successful, keeping a part of the tank volume empty to ensure local runoff control while reserving a portion for water supply.

Domestic RWH systems are generally of limited size, and excess runoff volumes can be allowed to overflow into an infiltration system, contributing to groundwater recharge. For stormwater control targets, it is suitable to integrate RWH with other sustainable drainage systems, such as green roofs and infiltration systems, to strengthen their individual effects [14,45].

For domestic RWH, any impervious surfaces, such as terraces, courtyards, and roads surrounding the building, can collect rainwater, but roofs are generally preferred. They usually contain low concentrations of chemical and microbiological pollutants requiring no or low treatment. Moreover, they have low hardness and salt concentrations and so meet the standards of most non-potable domestic uses.

Several authors estimated that rainwater could supply about 50% of the domestic water demand. Ghaffarian-Hoseieni [46] estimated that the percentage could achieve 80–90% of household water consumption. Toilet flushing alone can demand 60% of the rainwater collected by roofs and stored in a RWH system. The demand can be considered constant on a daily, weekly, and yearly scale, depending on the building and its use (residential, commercial, recreational). Irrigation can need about 16% of the water collected in a RWH system. Average annual volumes (specific for the unit of area) can range between 150 and 400 L/m$^2$, and depend on the type of vegetation and climate. In temperate climates, there is a seasonal fluctuation of water demand depending on vegetative periods. Washing machines can use 20% of rainwater collected by RWH. The water demand can be considered constant on a weekly scale. Other possible uses within buildings are car and surface washing, boiler supply, firefighting, thermal energy recovery, and building cooling [47]. The use of rainwater for drinking is generally disregarded in developed countries, even if modern technologies ensure high-quality standards [48]. Multiple water demands ensure relatively continuous use of rainwater, avoiding its stagnation and quality degradation issues.

RWH systems can range from a crude barrel to advanced systems composed of several integrated elements. The degree of modern implementation of RWH greatly varies within countries, and often systems do not maximize potential benefits [7]. Economic constraints and local regulations strongly influence the design of RWH systems.

Modern RWH systems include several integrated elements meeting high technological standards. They include roof, gutter protection screening, gutter, rain head, first flush diverter, tank screen, rainwater tank, insect proof flap valve, auto-fill system, pump system, irrigation filter, and water level indicator [49].

Roofs connected with a RWH system should be painted only with products approved for potable water, avoiding lead, zinc, copper products, and wood that absorbs moisture. Gutters should have protective screening to keep large debris from the pipework entrance. The installation of a rain head (downspout filter) that allows self-cleaning filtration can provide additional filtration. The first-flush diverter helps to prevent the first flush of contaminated rainwater from entering the tank [50]. Generally, the first flush is assumed to equal the first 3–5 mm of rainfall. Filters, tank screens, and insect-proof flap valves also protect rainwater quality.

The tank is the main element of a RWH system. It can be underground, on the ground, or on the building's roof. It is operated by gravity or pumping. The size and material mainly depend on rainwater use. Rain barrels (plastic or metal container of a few cubic meters capacity) are suitable for irrigation and runoff control in single-household residential buildings. Concrete cisterns of larger size are preferable for multi-purpose RWH systems. Although high-capacity storage tanks may increase the benefits of RWH, limited space can often prevent their installation. Tank materials should be opaque (for above-ground tanks) to avoid the exposure of stored water to sunlight to minimize algal growth. Outside tanks should be protected against freezing and algae flowering; underground tanks should consider the installation area (traffic, groundwater level, chemical, physical and mechanical characteristics of the ground). Other possible configurations are interrelated modular systems and collapsible tanks, gutter-based collection and storage, or other high-level,

low-energy systems [51]. Moreover, an auto-fill system can keep a minimum amount of water in the tank. This element is essential for rainwater tanks connected to automatic irrigation systems to prevent the pump from running dry. The pump system provides pressurized rainwater for easier distribution. The installation of an irrigation filter inline after the pump catches any large debris that may have passed through the pump. A water level indicator can help monitor water usage from the tank. Modern RWH systems can be equipped with sensors and data acquisition systems to improve automation and control for the optimal management of water resources.

The main variables governing the design and performance of RWH systems are rainfall regime, the area and type of the catchment surface, and water demand. The main hydrological variables are rainfall volume, duration, inter-event time, and the number of events in the selected period (day, month, season, year). The temporal scale used in the analysis is a function of the rainfall regime, and water use [51,52]. The amount of rainwater that can be harvested and reused is strictly related to the roof surface and coverage. The runoff coefficient, the percentage of rainwater conveyed to the tank, can generally range from about 0.9, for pitched roofs with tiles, to about 0.3 for intensive green roofs. Water demand can be considered continuous and constant, such as a daily request for toilet flushing, or discontinuous and variable, such as a monthly water request for irrigation. Rainwater reuse for domestic uses other than human consumption requires a dual distribution network internal to the building. RWH systems require regular maintenance, inspection, and cleaning of debris and sediment build-up, inlets/outlets/withdrawal devices, overflow areas, pumps, and filters.

Previous literature has used empirical relationships [53,54], stochastic analysis [55,56], or continuous mass balance simulation [52,57–59] to design RWH systems.

Simplified methods are suitable for systems of low-average sizes and have the average rainfall volume and water demand as inputs on an annual scale. The demand side and the supply side approach are the most well-known, belonging to this category. The first does not consider the rainfall regime and only considers the water demand; the second only considers the rainfall regime, neglecting the water demand [60].

Analytical methods are suitable for systems of medium-large size. The most famous calculation models are the yield after spillage (YAS) and yield before spillage (YBS) algorithms [57]. They implement a continuous simulation of a balance equation which considers the average values of the inflow, outflow, and water demand in the selected time scale. They estimate the ability of a tank to supply the water demand and can be used for design. The YAS algorithm overestimates the tank capacity, while the YBS underestimates it. The use of the YAS algorithm is suitable for design purposes, resulting in a more conservative estimate of the water demand supplied. The behavior of the tank is simplified by considering the yield only at the end of the selected time interval. To overcome this limit, Latham [61] introduced a further parameter into the algorithm to consider that the yield can occur during the event.

In recent decades, analytical–probabilistic approaches were proposed as an alternative to design event methods and continuous simulation for the design and modeling of RWH systems. They allow for the estimation of the probability distribution functions of the variables of interest to RWH design. Recently, such approaches have been applied to SuDS, such as infiltration systems [62,63], green roofs [64,65], and RWH [66,67]. Different targets, such as flood control, water supply, and environmental protection, can be included in the modeling [68]. The approach had the same reliability as continuous simulation [69]; moreover, it presents high flexibility when applying the technique to different climatic regimes and configurations.

User behaviors are often difficult to quantify; the stochastic nature and the high variability of the water demand depend on different uses and socio-technical factors (work patterns, household demographics, etc.). Behavioral-modeling tools can include multiple concurrent demand patterns (toilet flushing, garden irrigation, etc.) [70]. To ensure accurate outputs, high-resolution demand data may be required. Moreover,

sensitivity analyses are required where behavioral models use limited or uncertain data. When modeling RWH, the input variables' variability can be very significant. For this reason, low temporal resolution data (daily or sub-daily time steps) are preferred where possible. The ratio between the tank volume and the runoff volume gives indications about the temporal scale suitable for the analysis: if it is lower than 0.01, hourly or sub-hourly simulations are suggested; if it is about 0.125, the daily scale is preferred; if it is higher than 0.125, the monthly scale is suitable.

Aligning, in so far as possible, local rainfall patterns with water demand can substantially increase the efficiency of the RWH system in terms of water supply and conservation and stormwater control [71]. Several studies estimated the performance of RWH systems, collecting data from different locations and evaluating several scenarios in terms of tank size, irrigation area, and the number of people in the building [72,73]. Most consider toilet flushing, laundry, and irrigation as the main uses. The performance of the RWH also depends on the rainfall regime and the characteristics of the event, i.e., extreme events and long-duration storms [74].

Several indices try to assess the performance of RWH systems. Among these, the temporal indices measure the number of days in which the tank is not empty and/or the fraction of time in which the water demand is fully satisfied; the volumetric indices measure the efficiency in terms of yielded volume (over the water demand) and spilled volume (over the runoff volume). The ratio between demand and runoff volume gives an indication regarding the performance of the system: if it is lower than one, it means high performance even with a small tank; if it is equal to one, there is a significant variation in performance depending on tank volume; and if it is higher than one, it means low performance despite the tank size.

The performance of an RWH system mainly depends on the target to be achieved. Water saving and stormwater control are conflicting objectives of RWH systems, and different tank sizes may be needed to obtain the optimal benefit for each target. Mugume et al. [74] showed that RWH systems for UK houses could provide 95% of the users' non-potable water demand, maintaining sufficient attenuation capacity to control stormwater runoff for 1–100 year design storm. Palla and Gnecco [75], analyzed 144 different cases (considering four degrees of urbanization, three drainage network configurations, four precipitation regimes, and three return periods of the rainfall events), concluded that the effectiveness of an RWH system in supporting urban flood management becomes significant when storage tanks can contain at least 40% of the runoff volume. Morales-Pinzon et al. [76] showed through a life cycle analysis (LCA) that the introduction of environmental objectives (associated with emissions and the materials used) impact significantly tank size, depending on the type of building. To obtain reliable results and optimizing the performance of the RWH, multiple criteria analysis (MCA) is suggested.

Often, a benefit–cost ratio (BCR) analysis is also needed to support the feasibility of the technical solution identified by design methods. The balance between benefits and costs during the life cycle allows for the assessment of the financial viability of RWH systems. In the financial analysis, the water price is the primary variable, not only the actual price but also the expected one. The cost must include maintenance, operational, and energy consumption costs. Most approaches are simplified and do not holistically assess all potential benefits from RWH. The analysis should consider different climate conditions, building types, rainwater use, objectives, number of users, and tank size. Comparing results with those applied to traditional water supply systems (in terms of satisfying water demand) and urban drainage systems (in terms of stormwater control) allows for the estimation of the advantages of using RWH. Some environmental benefits, such as those related to environmental protection due to the reduction in combined sewer overflows and groundwater recharge, are complex to quantify. The same is true for indirect benefits, such as the reduction in the need to upgrade water infrastructure (water supply and urban drainage systems and wastewater treatment plants), energy saving, and emission reduction due to reduced water abstraction, transport, and treatment. Other benefits include the

increased agricultural efficiency of urban gardens and rural contexts and reduced flood costs [77]. Some studies [78,79] estimated climate change's influence on RWH systems' performance. They concluded that the reliability of the systems could slightly reduce in the future. Mainly, climate changes only marginally affect the dependability of RWH systems [36,80].

## 4. Rainwater Treatment

### 4.1. Quality of First-Flush Roof Runoff and Harvested Rainwater

Rainwater is considered a clean commodity, and its treatment methods have received significant interest in recent years. For domestic and industrial applications, the main concern is its quality characteristics. Compared to surface water and groundwater, rainwater has a nearly neutral pH, no hardness, and no presence of any disinfection by-products [81]. However, the physical, chemical and microbiological characteristics of first-flush roof runoff and harvested rainwater is highly affected by the catchment characteristics, storage material properties, and environmental conditions [82,83]. For instance, Despins et al. [84] reported that the rainwater collected on the catchment surfaces comprised of steel material adsorbs less atmospheric particulates than asphalt material and delivers higher-quality rainwater. They also reported that the pH of rainwater stored in plastic reservoirs tends to be marginally acidic. In contrast, the rainwater stored in concrete containers is slightly basic in nature. Regarding chemical quality, the presence of total organic carbon (TOC), nitrate ($NO_3^-$), and sulfate ($SO_4^{2-}$) is likely in rainwater due to the excrement of birds and rodents, lichens, and other depositions on the surfaces from which the runoff occurs. Compared to rooftops with concrete tiles, clay tiles, and galvanized steel material, roofs with wooden shingles are reported to promote the growth of lichens and mosses due to their relatively high porosity which subsequently increases the TOC, $NO_3^-$, and $SO_4^{2-}$ levels of the rainwater [85]. Concerning microbiological quality, the rainwater harvested from roofs covered with galvanized steel sheets was reported to have the lowest number of bacteria and adenosine triphosphate content compared to runoff from roofs covered with concrete tiles, ceramic tiles, and epoxy resin [86]. Additionally, Zdeb et al. [86] emphasized the effects of environmental conditions, particularly temperature, on the microbiological quality of the rainwater. They reported that the rainwater collected in the autumn tends to have the best microbial quality while water collected during summer has the worst.

Most studies which focus on the quality of the first-flush roof runoff and harvested rainwater suggest that the collected rainwater quality needs to be of higher quality for direct use [87] due to the microbiological quality risks owing to the presence of coliforms [88] and potential human pathogens such as Legionella and adenovirus [89].Thus, disinfection is necessary before rainwater use. Furthermore, the prevalence of heavy metals and other inorganic ions in rainwater due to fossil fuel combustion, and the likely presence of pesticide residues and fertilizers due to agricultural activities, [90] create a demand for advanced treatment strategies.

### 4.2. Rainwater Treatment State of the Art

Generally, rainwater is poor in biodegradability, so physicochemical treatment is a suitable option for improving rainwater quality for domestic and industrial use. There are many physicochemical treatment options available and selection is entirely determined by the required quality of the effluent and the recommended use of the treated water [91,92]. The treatment options generally proposed for rainwater can be divided into two broad categories: disinfection and filtration. In addition, recently proposed biological treatment options for rainwater treatment are also briefly discussed below.

#### 4.2.1. Disinfection

Due to the likely presence of pathogens, drinking untreated harvested rainwater could impact human health. Numerous techniques have been proposed to make the harvested rainwater meet potable water standards. Out of the many options, disinfection has received

the greatest attention [93]. Chlorination is the most widespread disinfection technique adopted in rainwater treatment systems, mainly due to its affordability [94]. Nonetheless, chlorination has several challenges, such as inconvenience in storing chemicals, taste and odor problems, and failure to eliminate microorganisms such as Cryptosporidium parvum and Giardia lamblia cysts [95]. Furthermore, to properly use chlorination, the dose and chlorine demand must be calculated by performing tests that might not be feasible at the household level. Although not as cost-effective, ultraviolet (UV) disinfection is also suggested as a potential alternative to chlorination for improving the microbiological quality of rainwater [96]. UV disinfection alters the DNA/RNA composition of pathogens and effectively destroys protozoa such as Cryptosporidium and Giardia. The advantage of UV disinfection over chlorination is that it does not generate any disinfection by-products. However, unlike chlorine treatment, UV disinfection produces no residual effect. Hence, recontamination of the treated rainwater can occur shortly after UV treatment if kept in a storage device [97].

Though old-fashioned, one of the most effective disinfection methods is raising the water temperature to 50–70 °C. At this temperature, the heat will either kill the pathogens or inactivate them [87]. Raising of temperature can be achieved either by boiling [98] or solar disinfection (SODIS) [99]. In addition to inducing heat, the portion of solar radiation (UV-A, 315–400 nm, visible violet, and blue light in the range of 400–490 nm) work synergistically in inducing microbicidal effects when using the SODIS technique [100]. The effectiveness of SODIS techniques in inactivating pathogens and other microorganisms and their ability to improve the microbiological quality of rainwater has been verified by several researchers [101,102]. The main disadvantage of the SODIS technique is that it does not offer any residual disinfection.

### 4.2.2. Filtration

Although pathogen removal is necessary to enhance the suitability of rainwater for domestic consumption, the effectiveness of disinfection techniques is influenced by the physical characteristics of rainwater, specifically, turbidity. Several filtration methods have been suggested, such as slow sand filtration, dual media filtration (sand, coal, and gravel), and membrane filtration [95]. Apart from removing the suspended solids from rainwater, filtration techniques induce other benefits, such as significantly improving physiochemical and microbiological characteristics, removing odor and taste problems, decreasing turbidity, and lowering the dose of chemicals for disinfection [103,104]. Below is an overview of the different filtration techniques recommended for rainwater treatment.

Ceramic filters are cone-shaped filters manufactured from locally acquired clay and are low-cost treatment options for rainwater [105]. During manufacture, the clay is mixed with rice husks and water to induce porosity and is painted with silver nitrate to reduce microbial growth. These filters have been reported to be effective in removing *E. coli* and other bacteria from rainwater [106]. However, these filters cannot treat large quantities of rainwater.

Compared to ceramic filters, the slow sand filter consists of sand and supporting gravel beds. The advantages of a slow sand filter includes low capital cost, straightforward design and construction, and low operational cost. The slow sand filters can be easily scaled up for small or medium towns or large villages. However, the large area requirement for large-scale applications makes them less attractive. It has been reported that slow sand filters are very effective in removing heavy metals, protozoa, *E. coli*, and bacteria from rainwater [107]. Furthermore, due to their ability to reduce turbidity to a significant extent (~95%), a slow sand filter also enhances the efficiency of disinfection techniques. Nonetheless, a slow sand filter may not function efficiently with a highly turbid water supply and is inefficient in odor removal [108]. Another commonly adopted filtration technique employs granular activated carbon (GAC) media instead of sand. These filters primarily work based on the principle of adsorption and are commonly used to remove natural organic matter, odor, and unpleasant taste [109]. Although there are reports of GAC filters removing turbidity, *E. coli*,

and total coliform [110], they are not considered effective in removing bacteria and viruses from rainwater [106], and turbidity can significantly reduce their lifespan. Dual media filters combine sand and GAC media and are commonly applied to remove *E. coli* from water. They are more effective than slow sand filters and have a longer lifespan. However, their performance is reported to vary with environmental conditions [111]. Furthermore, dual media filters are reported to be less efficient in removing turbidity than slow sand filters. Hence, additional coagulation–flocculation treatment may be required for treating highly turbid waters [112].

Traditional membrane filtration techniques include reverse osmosis (RO), nanofiltration (NF), ultrafiltration (UF), and microfiltration (MF), and are categorized based on pore diameter [113]. Out of the four membrane techniques, MF is traditionally the most utilized in rainwater treatment [114]. However, single MF units can only eliminate a fraction of large-diameter organics and pathogens, so they are not very effective as a rainwater treatment technique [115]. Thus, Shiguang et al. [116] recommend membrane surface modifications or combined use with other water treatments for effectively improving the performance of MF techniques in rainwater treatment. Nonetheless, considering the current advancements in MF techniques, there are ample reasons to believe that MF techniques have a future use for rainwater treatment [117].

Compared to MF techniques, UF techniques have higher filtration capacity and more significant potential for rainwater treatment. UF membranes also repel more macromolecular substances than MF membranes [114]. Therefore, these membranes can remove colloidal and/or suspended solids and pathogens in rainwater and yield demineralized effluent [118]. Compared to RO techniques, the production of demineralized effluent is less energy intensive and more cost-effective. Due to this reason, they are more environmentally friendly. However, in order to become a more attractive rainwater treatment technique, UF membranes must improve the filtration of greasy matter, increase the removal of heavy metals, and reduce the risk of membrane fouling [119]. Therefore, UF membrane technology still needs improvement in order to be used for practical applications in rainwater treatment.

NF membranes have properties in between UF and RO membranes, so utilizing NF techniques for rainwater treatment makes it viable to produce high-quality potable water [120]. In a recent study, Köse-Mutlu [121] has shown that when applied for rainwater treatment, NF techniques can achieve up to 99% natural organic matter removal and >99% $SO_4^{2-}$ removal. This proves the capability of NF membranes to remove organic and inorganic ions from water and effectively produce demineralized effluent. The evidence from another study [122] showed that NF membranes could reduce the opportunistic pathogens load (from 23.4% to 7.77%) and ensure the biosafety of the treated effluent. Yu et al. [122] also reported that the rainwater treated using NF techniques shows a lower disinfection by-product formation, making it safer for chlorine treatment. However, the effects of manufactured emerging contaminants were not considered in their study. Hence, further exploration is required to substantiate the claims of beneficial NF techniques for rainwater treatment in the literature.

RO techniques have been extensively researched in the field of rainwater treatment. The most well-known example is Singapore, which uses RO technology for rainwater treatment [123]. The RO membranes can effectively remove dissolved and colloid solids and opportunistic pathogens in rainwater by up to 99.9% [122]. Though RO techniques can produce high-quality effluent, it comes with considerable disadvantages. RO membranes demand frequent maintenance, and its absence can easily cause membrane fouling, leading to higher operating pressure, flux decline, and shorter membrane life [124]. Future research is expected to develop innovative membrane materials, enhance filtration efficiency, lower energy use, and alleviate membrane fouling [125–128].

4.2.3. Biological Treatment Options

Recently, biological treatment methods that facilitate the reduction in persistent organisms and the nonselective removal of microbial contaminants have gained attention.

Among them, biological treatments employing predatory bacteria and bacteriophages have received more attention. *Bdellovibrio* and like organisms are a group of Gram-negative bacteria identified as probable "live antibiotics" because of their ability to prey on and lower the concentration of primarily Gram-negative bacteria in co-culture experiments [129]. Waso et al. [130] applied *Bdellovibrio bacteriovorus* as a pretreatment to SODIS and solar photocatalysis for treating synthetic rainwater spiked with pathogens (*Klebsiella pneumonia* and *Enterococcus faecium*). The results showed that the pretreatment with *Bdellovibrio bacteriovorus* could effectively enhance the disinfection of Gram-negative bacteria in particular, such as Klebsiella and Enterococcus. However, the efficiency of predatory bacteria in disinfecting rainwater samples that contain mixed bacterial communities is yet to be investigated. In addition, the real-world applications of combining biological treatment constituting predatory bacteria with physical treatment methods are yet to be validated.

*Bacteriophages*, viruses that infect and lyse bacteria [131], have also been investigated for the targeted removal of pathogens from aquatic systems [92]. However, studies have reported that bacterial species may develop resistance to bacteriophages over time [132]. Hence, this must be addressed in order to apply bacteriophage in microbiological quality control of water samples successfully. Recently, Al-Jassim et al. [133] and Reyneke et al. [134] integrated bacteriophage treatment with SODIS to treat water samples. Results from the studies indicated the effectiveness of bacteriophage treatment. However, the efficiency of the bacteriophages for water treatment was only analyzed in small-scale experiments. The real-world functionality of bacteriophages in rainwater treatment is yet to be studied.

In addition to the above, bioretention is another popular rainwater management technique often employed in urban environments to deal with water quality issues. Bioretention systems consist of the vegetation at the top, followed by a substrate (growth media), drainage module, and an underdrain. Vijayaraghavan et al. [135] reported that although the advantages of bioretention systems for rainwater treatment are attractive from the environmental sustainability viewpoint, more concrete research studies are needed to ensure actual knowledge of the performance of these systems over an extended period of operation in the field.

### 4.2.4. Recent Trends

Recently, gravity-driven membrane (GDM) filtration processes have received significant interest from researchers as a new approach for water and wastewater treatment [136]. GDM filtration systems comprise a membrane filter (traditionally MF or UF), and water flows under gravity through the membrane in the dead-end filtration mode [114]. Research has shown that GDM filtration can achieve a continuous permeate flux at a relatively low gravity-driven force (40–100 mbar), without backwashing, flushing, or cleaning [137]. The flux stability in GDM systems is linked to the heterogeneous composition of the biofilm layers which form over the membrane surface [138,139]. These improvements of GDM systems over the traditional membrane filtration techniques result in a more cost-effective treatment option for rainwater. A recent study has verified the prospect of applying a GDM system for rainwater treatment by coupling the membrane bioreactor with an electrocoagulation unit [140]. Their findings show that the GDM system can effectively remove turbidity, ammonia-N, total phosphorous, heavy metals, and natural organic matter, and produce high-quality effluent under varying operating conditions. Though the outcomes are promising, further exploration is required to validate the effectiveness of GDM systems for rainwater treatment under field conditions.

### 5. Trends and Perspectives

Although the benefits of RWH have been well documented, their implementation is somewhat sporadic [7]. Recent trends have focused on addressing this by demonstrating the multi-purpose nature of RWH in terms of its environmental, financial, and social benefits. Quinn et al. [34] address this by suggesting a framework incorporating water supply and stormwater management metrics that provide a robust characterization of

performance during significant rainfall events and on a longer-term basis. However, currently, this framework does not contain any of the societal and economic benefits which are more difficult to quantify. The design of these systems can also act as a barrier to implementation; for example, in the UK, these systems are designed to manage runoff from 1 in 100-year design storms, which results in recommendations for large, costly tanks [141]. Stovin et al. [141] apply their framework to design, illustrating that a balance between size and stormwater management performance can be achieved by designing for the retention of rainfall events with smaller return periods. Recently, larger-scale modelling has been applied to demonstrate the utility of advanced technology, such as real-time control (RTC), to RWH systems. For example, Xu et al. [142] illustrate the benefits of RTC of RWH on reducing erosion and restoring the pre-development conditions in sensitive receiving waters and suggest that investments in RTC technology would appear to be more promising than investments in increasing RWH detention volume.

Campisano et al. [7] highlighted that financial viability is a significant barrier to implementation. As such, LCA has been adopted to examine the environmental and economic costs of RWH. It is challenging to compare different LCAs due to the assumptions made when creating them and their sensitivity to geographical parameters. Leong et al. [143] compared decentralized RWH, greywater recycling, and hybrid rainwater–greywater systems and found RWH to be the optimal option, as it had the second highest mains water savings, lowest environmental impact scores relative to mains water in seven categories (i.e., acidification, eutrophication, freshwater ecotoxicity, global warming, human toxicity, photochemical ozone creation, and water stress index), and is the first system to become financially attractive at USD 2.00/m$^3$. Ghimire et al. [144] found similar results with their RWH system outperforming the mains water system in all categories except ozone depletion, although they did not examine cost. Van Dijk et al. [145] apply a different approach to illustrating the financial benefits of RWH; they demonstrate that well-designed and implemented rooftop RWH systems can meet multiple infrastructure development needs of a city with reduced public expenditure as compared to centralized systems, and that RWH is a viable, profitable climate change adaptation strategy. One of the key challenges when planning infrastructure is the uncertainty regarding future climate scenarios. As discussed throughout this paper, RWH is a viable option to mitigate the impacts of both drought and floods; however novel approaches to value this flexibility are needed. Deng et al. [146] and have already proposed a framework to appraise investments in urban water management systems under uncertainty. Following on from this work, Manocha and Babovic [147] add a cost–benefit analysis to decision-making approaches focusing on uncertainty, which provides additional insights to policymakers.

The decentralized nature of many RWH systems offers a unique opportunity for communities to be actively involved in water management, which has been shown to yield multiple benefits [148]. RWH is often used as part of a systematic catchment-based approach to stormwater management, where multiple SuDS are used to holistically manage surface water runoff. For example, the sponge city initiative in China has championed this approach and investigations into the optimal placement of systems to manage urban flooding is ongoing [149]. Sefton et al. [28] suggest there are transformative advantages to a more community-oriented approach to flood resilience by including participatory RWH management, particularly the move towards a process of mutual learning and two-way communication.

## 6. Conclusions

This paper provides an overview of recent developments and trends in the field of rainwater harvesting. It shows that the advantages of RWH are often understated and there is potential to link this practice to all of the SDGs; however, the limitations of the systems and current research is acknowledged. Regulations are of great importance in ensuring the widespread implementation of this systems and in the future, they should not only include technical and environmental guidance but also economic and

social supports. Similarly, trends in the advancement of RWH research towards more multidimensional benefit analysis are shown. Modern RWH systems include several components and high-level technologies. The design of the tank is crucial to satisfy different water needs and the variability of stormwater patterns must be considered. Multi-objective design of tanks is needed to increase the reliability and the efficiency of the systems to meet different goals.

In addition, this paper examined the existing state of the art in rainwater treatment. The characteristics of different physicochemical treatment options in rainwater treatment, specifically, disinfection and filtration, with emphasis on membrane technologies, were summarized. The recent developments in biological treatment options for rainwater treatment were also analyzed. The research on GDM techniques and the process of treating rainwater with various physicochemical and biological technology combinations are still under analysis. The current advancements in the state of the art prove that future prospective treatment techniques are worth looking forward to.

**Author Contributions:** Conceptualization, A.R., R.Q. and A.O.; formal analysis, A.R. and R.Q.; investigation, A.R. and R.Q.; writing—original draft preparation, A.R., R.Q. and G.R.A.; writing—review and editing, A.R., R.Q. and G.R.A.; supervision, A.O., A.R. and G.B. All authors have read and agreed to the published version of the manuscript.

**Funding:** This research received no external funding.

**Data Availability Statement:** No new data were created or analyzed in this study. Data sharing is not applicable to this article.

**Conflicts of Interest:** The authors declare no conflict of interest.

## Nomenclature

| | |
|---|---|
| BCR | Benefit Cost Ratio |
| BMP | Best Management Practices |
| GAC | Granular Activated Carbon |
| GDM | Gravity Driven Membrane |
| LCA | Life Cycle Assessment |
| LID | Low-Impact Development |
| MCA | Multiple Criteria Analysis |
| MF | Microfiltration |
| NF | Nanofiltration |
| RO | Reverse Osmosis |
| RTC | Real-Time Control |
| RWH | Rainwater Harvesting |
| SDG | Sustainable Development Goal |
| SODIS | Solar Disinfection |
| SuDS | Sustainable Drainage Systems |
| UF | Ultrafiltration |
| UN | United Nations |
| UV | Ultraviolet |
| YAS | Yield After Spillage |
| YBS | Yield Before Spillage |
| WMO | World Meteorological Organization |

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
