# Peer review of "Rainwater Harvesting and Treatment: State of the Art and Perspectives"

_water, doi:10.3390/w15081518_

Round 1

Reviewer 1 Report

This review paper provides a reasonably comprehensive overview of rainwater harvesting systems. The contribution discusses harvesting and treatment of rainwater, in the context of sustainability goals as well as economic assessment.

I am somewhat surprised that bioremediation is not discussed to a greater degree. Bioremediation is particularly significant since it offers water quality treatment in a low-energy regime. It may be of interest to incorporate one more review paper that focuses on stormwater management systems:

K Vijayaraghavan et al 2022, Bioretention systems for stormwater management: Recent advances and future prospects, Journal of Environmental Management 292, 112766

Furthermore, I would like to encourage authors to consider discussing systemic perspectives and the significance of catchment-scale implementation of rainwater harvesting. 

P Schmitter et al, 2016, Effect of catchment-scale green roof deployment on stormwater generation and reuse in a tropical city

Finally, one of the hallmarks of RWHS is the ability to cope with uncertain future scenarios. This is particularly important from the perspective of climate change. Hence, the valuation of RWHS in the context of stormwater management requires distinctly novel approaches. The two papers below introduce specific methodologies to value flexibility as an intrinsically significant characteristic of the RWH systems:

Y Deng et al, 2013, Valuing flexibilities in the design of urban water management systems, Water Research 47 (20), 7162-7174

N Manocha and V Babovic, 2017, Development and valuation of adaptation pathways for stormwater management infrastructure, Environmental Science & Policy 77, 86-97

Author Response

We would like to thank the author for taking the time to review our paper and for their insightful comments and suggestions. We have responded to their comments in bold below.

I am somewhat surprised that bioremediation is not discussed to a greater degree. Bioremediation is particularly significant since it offers water quality treatment in a low-energy regime. It may be of interest to incorporate one more review paper that focuses on stormwater management systems:

K Vijayaraghavan et al 2022, Bioretention systems for stormwater management: Recent advances and future prospects, Journal of Environmental Management 292, 112766

To acknowledge the biological treatment aspects of rainwater, a new section titled 4.2.3. Biological treatment options has been added (see below). We hope the modifications are in order with the expectations of the reviewer.

‘Recently biological treatment methods that facilitate the reduction of persistent organisms and the nonselective removal of microbial contaminants have gained attention. Out of them, biological treatment employing predatory bacteria and bacteriophages has received more favor. Bdellovibrio-and-like organisms are a group of Gram-negative bacteria identified as probable “live antibiotics” because of their ability to prey on and lower the concentration of primarily Gram-negative bacteria in co-culture experiments [130 ]. Waso et al. [131] applied Bdellovibrio bacteriovorus as a pretreatment to SODIS and solar photocatalysis for treating synthetic rainwater spiked with pathogens (Klebsiella pneumonia and Enterococcus faecium). The results showed that the pretreatment with Bdellovibrio bacteriovorus could effectively enhance the disinfection of particularly Gram-negative bacteria such as Klebsiella and Enterococcus. However, the efficiency of predatory bacteria in disinfecting rainwater samples that contain mixed bacterial communities is yet to be investigated. In addition, the real-world applications of combining the biological treatment constituting predatory bacteria with physical treatment methods are yet to be validated.

Bacteriophages, viruses that infect and lyse bacteria [1 32], have also been investigated for the targeted removal of pathogens from aquatic systems [93]. However, studies have reported that bacterial species may develop resistance to bacteriophages over time [1 33]. Hence, this must be overcome to apply bacteriophage in microbiological quality control of water samples successfully. Lately, Al- Jassim et al. [134 ] and Reyneke et al. [1 35] integrated bacteriophage treatment with SODIS to treat water samples. Results from the studies indicated the effectiveness of bacteriophage treatment. However, the efficiency of the bacteriophages for water treatment was only analyzed in small-scale experiments. The real-world functionality of bacteriophages in rainwater treatment is yet to be studied.

In addition to the above, bioretention is another popular rainwater management technique often employed in urban environments to deal with water quality issues. Bioretention systems consist of the vegetation at the top, followed by substrate (growth media), drainage module, and an underdrain. Vijayaraghavan et al. [1 36] reported that although the advantages of bioretention systems for rainwater treatment are attractive from the environmental sustainability viewpoint, more concrete research studies are needed to ensure actual knowledge of the performance of these systems over an extended period of operation in the field.’

Furthermore, I would like to encourage authors to consider discussing systemic perspectives and the significance of catchment-scale implementation of rainwater harvesting.’

P Schmitter et al, 2016, Effect of catchment-scale green roof deployment on stormwater generation and reuse in a tropical city

To acknowledge systemic perspectives at a catchment scale, the following has been added to 5. Trends and Perspectives

‘RWH is often used as part of a systematic catchment-based approach to stormwater management, where multiple SuDS are used to holistically manage surface water runoff. For example, the sponge city initiative in China has championed this approach and investigations into the optimal placement of systems to manage urban flooding is ongoing [150].

Tansar, H.;Duan, H; Mark, O. Catchement-Scal and Local Scale Based Evaluation of LID Effectiveness of Urban Drainage System Performance. Water Resour. Manag. 2022, 36, 507-526. doi: 10.1007/s11269-021-03036-6

Finally, one of the hallmarks of RWHS is the ability to cope with uncertain future scenarios. This is particularly important from the perspective of climate change. Hence, the valuation of RWHS in the context of stormwater management requires distinctly novel approaches. The two papers below introduce specific methodologies to value flexibility as an intrinsically significant characteristic of the RWH systems:

Y Deng et al, 2013, Valuing flexibilities in the design of urban water management systems, Water Research 47 (20), 7162-7174

N Manocha and V Babovic, 2017, Development and valuation of adaptation pathways for stormwater management infrastructure, Environmental Science & Policy 77, 86-97

The following has been added to the 5. Trends and Perspectives.

‘One of the key challenges when planning infrastructure is the uncertainty regarding future climate scenarios. As discussed throughout this paper, RWH is a viable option to mitigate the impacts of both drought and floods, however novel approaches to value this flexibility are needed. Deng et al. [147] and have already proposed a framework to appraise investments in urban water management systems under uncertainty. Following on from this work, Manocha and Babovic [148] adds a cost-benefit analysis to decision-making approaches focusing on uncertainty, which provides additional insights to policymakers.’

Reviewer 2 Report

A well organized review of rainwater harvesting systems.

Author Response

We would like to thank the reviewer for taking the time to review our paper.

Reviewer 3 Report

The review by the authors from Italy is concerned with "Rainwater Harvesting and Treatment: State of the Art and Perspectives".

Thank you for the opportunity to review such a useful and actual paper.

The manuscript presents a review of the state of the art of rainwater harvesting, treatment, and management. It focuses on the environmental and social benefits of rainwater harvesting and its technical and acceptance limits.

This submitted article could be within the aim and scope of the MDPI journal Water; submitted to Section: Urban Water Management,

https://www.mdpi.com/journal/water/sections/Urban_Water_Management

Rainwater Harvesting and Treatment

https://www.mdpi.com/journal/water/special_issues/K9B9V1H636

  • Abstract and introduction: These two parts are focused on the paper's main aim and the new contributions of the authors to the state of the art. The abstract with keywords very effectively summarizes the manuscript.

The key objective for the authors is to show the well-known solution from history with different levels of advanced technology.

  • Materials and methods: Based on the existing research, the authors quantitatively showed the interest of the topic.

Their review discusses laws and regulations in different countries that encourage this practice. It presents also methodologies for designing a rainwater harvesting system and describes the influence of its variables and the temporal and spatial scale on the system's performance. The manuscript analyzes the most advanced technologies for rainwater treatment, too.

  • Results and discussion: The data are well presented with relevant and current tables, figures, and references. Authors introduce further developments on the topic, trends, and perspectives to increase rainwater harvesting, reuse, and management of rainwater harvesting.

With some adaptation, this review could be used more generally by other researchers.

Author Response

We would like to thank the author of the report for taking the time to review our paper.  We have made several changes which we believe have improved the document, such as the addition of a conclusion section, the addition of further information regarding systematic approaches to catchment management and the evaluation of systems' performance under uncertain future scenarios.

Reviewer 4 Report

The paper was submitted in tracking change form and it is difficult to understand that is the final form to be submitted to publication.  

The paper presents a pretty weak literature review of topics related to Rainwater Harvesting, without significant scientific analysis.

Author Response

We thank the reviewer for taking the time to read our paper and have attached a clean version for you to look at. We believe the paper makes a valuable contribution to this field of research as

  • It outlines various benefits of RWH and novelly links them to all SDGs using examples in a manner that has not previously been done.
  • It identifies characteristics of rules and regulations that are successful in increasing the implementation of these systems and suggests the inclusion of both economic and social supports, and links this to the pillars of sustainability. Again this is a unique approach to this issue.
  • The characteristics of different physicochemical and biological treatment options in rainwater treatment have been summarized. The review is expected to open fresh insights into various processes of treating rainwater by combining diverse physicochemical and biological technology options that are still under analysis.

Reviewer 5 Report

The manuscript presents a review on rainwater harvesting schemes across the world. The review article covers a little bit of everything regarding RWH, and is generally well written. But, as authors themselves stated, there is already a lot of literature on this topic, and it is not clear what is the novelty of this work. You highlight that there are a lot of RWH manuscripts published, but you do not show why yours is any different than all the others. Also, since the review in this work is not in-depth, but more descriptive in nature, a lot of the topic that this review covers are general knowledge at this point in time. While this manuscript presents a great teaching and industry publication, it lacks scientific novelty and rigour.

Some other general comments:

C1: Define terms “rainwater” and “stormwater”. You sometimes use these terms as synonyms (e.g. LN 244), but they are typically different types of water source, with rainwater usually considered as roof runoff and direct atmospheric water, while stormwater is road and paved surface runoff which is much more polluted with organic and inorganic pollutants. With this in mind, re-read the manuscript and use appropriate terminology.

C2: Section “Regulations and Laws” generalises world-wide standards, but doesn’t go into the specifics about water storage requirements, water quality requirements, types of uses, etc.

C3: It would be good to provide a Conclusions section, to finish with some general concluding remarks. At the moment, it feels abrupt.

Author Response

We would like to thank the author of the report for taking the time to review our paper and for their valuable comments. Our responses to your comments are given in bold below.

It is acknowledged that there are many papers on RWH available, but we believe our paper makes a valuable contribution to the field of research.

  • It outlines various benefits of RWH and novelly links them to all SDGs using examples in a manner that has not previously been done.
  • It identifies characteristics of rules and regulations that are successful in increasing the implementation of these systems and suggests the inclusion of both economic and social supports, and links this to the pillars of sustainability. Again this is a unique approach to this issue.
  • The characteristics of different physicochemical and biological treatment options in rainwater treatment have been summarized. The review is expected to open fresh insights into various processes of treating rainwater by combining diverse physicochemical and biological technology options that are still under analysis.

Some other general comments:

C1: Define terms “rainwater” and “stormwater”. You sometimes use these terms as synonyms (e.g. LN 244), but they are typically different types of water source, with rainwater usually considered as roof runoff and direct atmospheric water, while stormwater is road and paved surface runoff which is much more polluted with organic and inorganic pollutants. With this in mind, re-read the manuscript and use appropriate terminology.

The terminology of the manuscript has been revised to properly differentiate between stormwater and rainwater.

C2: Section “Regulations and Laws” generalises world-wide standards, but doesn’t go into the specifics about water storage requirements, water quality requirements, types of uses, etc.

The purpose of this section was not to review all laws and regulations as this has been done multiple times, but to identify best practices and link this offer guidance on how laws and regulation can be improved in the future.  Guidance is also given to the reader on where to obtain more through reviews.

To better reflect this purpose the abstract has been amended to replace

‘The review discusses primary laws and regulations in different countries that encourage this practice.’

With

‘The review identifies laws and regulations that encourage this practice and their current limitations.’

C3: It would be good to provide a Conclusions section, to finish with some general concluding remarks. At the moment, it feels abrupt.

Considering the reviewer’s comment, a new Conclusions section has been added.

‘6. Conclusions

This paper provides an overview of recent developments and trends in the field of rainwater harvesting. It shows that the advantages of RWH are often understated and there is potential to link this practice to all of the SDGs, however the limitations of the systems and current research is acknowledged. Regulations are of great importance in ensuring the widespread implementation of this systems and in the future they should not only include technical and environmental guidance but also economic and social supports. Simarlarly, trends in the advancement of RWH research towards more multidimensional benefit analysis are shown.

In addition, this paper examined the existing state-of-the-art rainwater treatment. The characteristics of different physicochemical treatment options, specifically disinfection, and filtration with emphasis on membrane technologies, in rainwater treatment were summarized. The recent developments in biological treatment options for rainwater treatment were also analyzed. The research on GDM techniques and the process of treating rainwater with various physicochemical and biological technology combinations are still under analysis. The current advancements in the state-of-the-art prove that the future application prospects are worth looking forward to.’

Round 2

Reviewer 5 Report

While you stated some gaps that this work is addressing in response to reviewer's comments, you have not edited article to highlight this novelties in the Introduction. You need to clearly present the innovation in the Introduction section (and Abstract) so that future readers are aware before reading the rest of the paper. 

Author Response

We would like to thank the reviewer again, for taking the time to review our paper.  We have amended the abstract and introduction to highlight our contribution to the field of research (see below)

Abstract

‘This study is aimed at reviewing the state of the art of rainwater harvesting, treatment, and management. It focuses on the environmental and social benefits of rainwater harvesting and novelly links them to the Sustainable Development Goals. The review identifies characteristics of laws and regulations that encourage this practice and their current limitations. It presents methodologies to design a Rainwater Harvesting (RWH) system, describes the influence of its design variables, and the impact of temporal and spatial scales on the system’s performance. The manuscript also analyzes the most advanced technologies for rainwater treatment, providing insights into various processes by combining diverse physiochemical and biological technology options that are in the early stages of development’

Introduction

‘This manuscript aims to review papers on rainwater harvesting and treatment and identify the most important findings and progress in this field of research.

Section 2, "Rainwater Harvesting", introduces the multiple advantages of this practice and its technical, social, and financial limitations (paragraph 2.1) and links them to the Sustainable Development Goals (SDGs), providing new insights into the utility of these systems. Paragraph 2.2 presents key characteristics of regulations, laws, and design manuals which can help or hinder widescale RWH implementation.

Section 3, "Rainwater Harvesting Systems", describes modern rainwater harvesting systems with a focus on domestic usage. It reviews the state of the art of methods and models for their design and modeling. It also deepens the performance of the systems depending on the different variables affecting the process, the spatial and temporal scale, and the target of rainwater harvesting.

Section 4, "Rainwater Treatment", presents current and emerging technologies to achieve set water quality standards for the different reuses of rainwater. ‘

Round 3

Reviewer 5 Report

X

Author Response

Thank you for taking the time to review our paper, it appears you have no further comments. If this is a mistake, please let some one from the editorial team know and we would be happy to address any further comments you have.